# A Summary of In Vitro and In Vivo Studies Evaluating the Impact of E-Cigarette Exposure on Living Organisms and the Environment

**DOI:** 10.3390/ijms21020652

**Published:** 2020-01-19

**Authors:** Anna Merecz-Sadowska, Przemyslaw Sitarek, Hanna Zielinska-Blizniewska, Katarzyna Malinowska, Karolina Zajdel, Lukasz Zakonnik, Radoslaw Zajdel

**Affiliations:** 1Department of Economic Informatics, University of Lodz, 90-214 Lodz, Poland; katarzyna.malinowska@uni.lodz.pl (K.M.); lukasz.zakonnik@uni.lodz.pl (L.Z.); radoslaw.zajdel@uni.lodz.pl (R.Z.); 2Department of Biology and Pharmaceutical Botany, Medical University of Lodz, 90-151 Lodz, Poland; przemyslaw.sitarek@umed.lodz.pl; 3Department of Allergology and Respiratory Rehabilitation, Medical University of Lodz, 90-725 Lodz, Poland; hanna.zielinska-blizniewska@umed.lodz.pl; 4Department of Medical Informatics and Statistics, Medical University of Lodz, 90-645 Lodz, Poland; karolina.smigiel@umed.lodz.pl

**Keywords:** e-cigarettes, e-liquid, oxidative stress, inflammation, lung pathologies

## Abstract

Worldwide use of electronic cigarettes has been rapidly expanding over recent years, but the long-term effect of e-cigarette vapor exposure on human health and environment is not well established; however, its mechanism of action entails the production of reactive oxygen species and trace metals, and the exacerbation of inflammation, which are associated with potential cytotoxicity and genotoxicity. The present study examines the effects of selected liquid chemicals used in e-cigarettes, such as propylene glycol/vegetable glycerin, nicotine and flavorings, on living organisms; the data collected indicates that exposure to e-cigarette liquid has potentially detrimental effects on cells in vitro, and on animals and humans in vivo. While e-liquid exposure can adversely influence the physiology of living organisms, vaping is recommended as an alternative for tobacco smoking. The study also compares the impact of e-cigarette liquid exposure and traditional cigarette smoke on organisms and the environmental impact. The environmental influence of e-cigarette use is closely connected with the emission of airborne particulate matter, suggesting the possibility of passive smoking. The obtained data provides an insight into the impact of nicotine delivery systems on living organisms and the environment.

## 1. Introduction

Although their initial inception dates back to 1963, the design of e-cigarettes in 2003 by Hon Lik, has spurred increasing global sales, which now show profits worth billions of dollars. Their popularity among the smoking community has also grown exponentially to the extent that e-cigarettes have become a leading product among the various alternatives to traditional cigarettes [1,2,3,4]. They comprise a battery-charged system containing a power source, a heater and a cartridge containing a liquid which transforms into vapor. The vapor produced from the liquid, usually containing propylene glycol (PG)/vegetable glycerin (VG), nicotine (N) and flavorings (F) [2], is inhaled by the user. By contrast, a list of chemical compounds in conventional cigarettes and cigarette smoke includes more than 7000 items of which 69 are known carcinogens [5]. E-cigarettes are widely regarded as a safer alternative to smoking, a known cause of significant morbidity and mortality accounting for more than seven million deaths annually [6]. Although they are also considered as a tool for harm reduction and smoking cessation [7,8], the effects of both short-term and long-term exposure remain largely unknown.

While smoking cigarettes is a well-known and major risk factor for lung cancer [9], recent studies suggest that e-cigarette smoking may also induce the signaling pathways associated with tumor development. Carcinogenesis, a process characterized by an abnormal proliferation of cells, involves several stages: its onset is characterized by genetic alteration and tumor initiation, which is followed by accelerated division of cells, leading to the outgrowth and accumulation of abnormalities associated with the unregulated cellular signaling pathways [10]. Lung cancer is both the most common cancer (2.1 million cases) and the most common cause of cancer death (1.8 million deaths) worldwide [11]. Its etiology is based on the occurrence of mutations in oncogenes leading to excessive proliferation of mutated cells and tumor formation [12,13]. One risk factor increasing the possibility of lung cancer is chronic obstructive pulmonary disease (COPD) [14,15], a progressive aggravation of lung function [15]. Exposure to e-cigarette vapor may trigger inflammation in the airways, which is a risk factor for the development of lung cancer; both COPD and lung cancer share an etiology of oxidative stress associated with inflammation [14]. Reactive oxygen (ROS) and nitrogen species (RNS) induce endothelial dysfunction and tissue injury by stimulating the production of chemical mediators of inflammation [16]. In addition, ROS also upregulate the expression of numerous genes involved in immune and inflammatory processes by activating nuclear factor κB (NF-κB), a member of the transcription factor protein family [17].

This review examines the literature concerning the mechanisms of action of e-cigarettes, the effect of exposure to e-liquid in vitro and in vivo and the environmental impact of electronic nicotine delivery system.

## 2. The Mechanisms of E-Cigarette Action

A typical constituent of e-liquid is nicotine. Consumption is known to increase the risk of adverse cardiovascular events, and absorbed nicotine may release catecholamine by the activation of nAChRs localized on sympathetic nerve endings; subsequent activation in vascular smooth muscle cells contracts the vascular tissues and elevates blood pressure. Nicotine may also exert a direct effect on vascular smooth muscle cells to induce vascular relaxation or contraction, thus altering cardiovascular function, both directly and indirectly via nicotine-induced adrenergic stimulation. The cellular mechanisms of nicotine action may be related to its ability to prolong action potentials and depolarize membrane potential [18]. However, the electronic nicotine delivery system poses a lower cardiovascular risk than tobacco smoking [19]. 

E-cigarette liquid formulations also contain a wide variety of flavoring agents [20]. Flavors may influence nicotine absorption by altering pH [21]. Although none have been tested so far, viz. acetylpyridine, diacetyl, dimethylpyrazine, cinnamaldehyde, eucalyptol, eugenol, isoamylacetate, menthol and vanillin, have been found to cause cultured endothelial cell dysfunction (porcine aortic endothelial cells) [22]; studies on cellular models suggest that they may exert a toxic effect on respiratory cells based on oxidative stress and inflammatory processes [23,24,25,26,27,28]. Contact with such flavorant agents results in elevated level of ROS production and excessive IL-8 secretion in airway epithelium cells, with both mechanisms leading to cell cytotoxicity: oxidative stress is a cause of DNA damage [29]. In both cases, various molecular cell signaling cascades are activated, followed by inflammatory responses in lung cells [30].

ROS are highly reactive chemicals possessing unpaired electrons in the outer shell. They can initiate oxidative stress followed by other pathological processes, and have been found to damage proteins, lipids and DNA and sustain proinflammatory responses. E-cigarette vapor has been found to contain 7 × 10^11^ free radicals per puff compared to 10^14^ in cigarette smoke [31,32]. ROS production has been found to be provoked by the inhalation of toxic agents, and to stimulate inflammatory signaling pathways [33]. One such stimulant of ROS release is exposure to the vapor from e-cigarettes [34,35]. Although biological targets produce antioxidant enzymes and small molecules to protect against ROS, analyses of lung epithelial cells in smokers have indicated that exposure to e-liquid vapor results in DNA damage and genotoxicity. These mechanisms may be connected with the suppression of cellular antioxidant defense [36]. ROS are released by human lung epithelial cells and fibroblasts (HFL-1) upon exposure to such flavoring compounds [37,38,39]. 

Inflammation is considered a primary cause of various types of cancer [40], including lung cancer [41] and COPD [42]. A key role of the immune system is to maintain cell and tissue homeostasis; however, carcinogenesis is characterized by a shift in this equilibrium from anti-inflammatory to proinflammatory pathways. Excessive production of key inflammatory agents and crucial cytotoxic mediators such as ROS, promote genomic instability, neoplastic development, tumor growth, metastasis and influence to response to chemotherapy. Such inflammation may be triggered by the constituents of e-liquid, which can be converted to propylene oxide, acrolein, acetaldehyde and formaldehyde upon inhalation. It has been found that a possible relationship may exist between exposure to e-liquid vapor and a proinflammatory effect on cells [43]. Some flavoring compounds, such as acetoin, ortho-vanillin, and maltol, increase IL-8 inflammatory cytokine secretion in the cells (Beas2B and HFL-1) [37].

Vaporized e-liquid solution is contaminated with various metals, such as aluminum, arsenic, cadmium, chromium, copper, iron, manganese, nickel, lead and zinc, which pose a threat to human health [44]. In addition, trace metals can also be delivered to the aerosol through components of the electronic nicotine delivery system, such as the atomizer and batteries; this contamination may be exacerbated by the high temperatures involved in the production of the aerosol [45]. Part of the heavy metals present in vapor, such as lead, are derived from *Nicotiana tabacum*, the source of nicotine in tobacco, which absorbs pollutants from the environment during its growth [46]. Trace metals can accumulate in smokers following prolonged use, thus resulting in adverse health effects. Studies suggest a close relationship between exposure to aluminum [47], arsenic [48], cadmium [49], chromium [50], copper [51], manganese [52], nickel [53], lead [54] and zinc [55] and predisposition to the following respective lung pathologies, diffuse parenchymal lung disease, impaired lung function, lung cancer, chronic lung inflammation and injury, pulmonary inflammation, occupational asthma, respiratory tract cancer, higher frequency of respiratory symptoms, altered regulation of the epithelial–mesenchymal transition (EMT) process. 

All the above processes may have a detrimental effect on the lungs, disrupt airway tract homeostasis and increase the risk of various lung pathologies and carcinogenesis.

## 3. The Effects of Exposure on E-Liquid Ingredients

The mixture used in e-cigarettes, e-liquid, is composed of three main ingredients: PG/VG, N and F. While all e-liquid components are generally recognized as safe by the Food and Drug Administration (FDA) [56], such they may become hazardous to health, particularly to the lungs, when heated and inhaled.

The main component of e-liquid, constituting around 89–90% of its formulation, is PG and/or VG. When subjected to high temperatures, it leads to the formation of carbonyl compounds, chemicals characterized by acute inhalational toxicity and irritant properties. PG and VG have been found to be significant sources of formaldehyde, acetaldehyde, and acrolein [57]. Exposure to formaldehyde exerts a negative effect on the respiratory system, causes airway irritation and impairs pulmonary function [58]. Acetaldehyde indirectly affects genome function via the formation of adducts with histone proteins, or by influencing the expression of histone acetyltransferase in human lung bronchial epithelial cells [59]. Acrolein plays a role in the induction of pulmonary inflammation and ROS generation, which is considered to be responsible for the inflammatory reaction; this compound may also contribute to DNA tumor suppressor gene p53 adduction, has been linked to mutations, and may be a key factor associated with the lung injury, COPD, lung cancer and asthma [60]. Although numerous studies suggest that exposure to the aforementioned chemicals, including formaldehyde, acrolein and acetaldehyde is related to the pathogenesis of asthma [61,62,63,64], Lechasseur et al. do not report any inflammatory responses in mice exposed to e-cigarettes PG vapor for two hours per day for eight weeks [65]. Additionally, Cotta indicates that PG does not appear to display toxicity effects via inhalation and does not regard it as a significant hazard [66].

Another important component of e-liquid vapor is nicotine, a highly addictive compound whose concentration ranges from 0 mg to 36 mg/mL [67]. Nicotine exerts its effects on the brain by modulating the activity of certain ligand-gated ion channels known as nicotinic acetylcholine receptors (nAChRs) [68]; these comprise junctions of individual subunits activated by the binding of endogenous neurotransmitter acetylcholine (Ach) and other biologically active compounds, including nicotine. Nicotine itself poses numerous health hazards by influencing several processes, such as cell proliferation, apoptosis and the immune response; however, it also contributes to oxidative stress and consequent DNA mutations, which can lead to cancer. Nicotine exposure is associated with an increased probability of tumor proliferation, metastasis, and can also increase the likelihood of treatment failure by reducing the effectiveness of chemotherapy and radiotherapy [69]. Nicotine promotes lung cancer development and accelerates its proliferation, angiogenesis, migration, invasion and EMT, via its influence on nAChRs receptors, whose presence has been confirmed in lung cancer cells [70].

Nicotine is predominantly metabolized in vivo into cotinine [71]; although its serum level constitutes a predictor of lung cancer risk [72], it is nevertheless believed to be nontoxic and noncancerogenic [73]. Nicotine is also metabolized into nitrosamines, chemicals with proven carcinogenicity, which lead to the occurrence of methylation intermediates in DNA [74]. It has been established that e-cigarettes induce O6-methyl-deoxuguanosine adducts and aldehyde-derived cyclic 1,N2-propano-dG adducts in lung, heart, liver, and bladder tissues in a mouse model, with the lungs demonstrating higher levels of adducts than the bladder or heart [75]. The covalent binding of carcinogens to DNA is believed to contribute to the induction of tumorigenesis by preventing DNA repair [76]. Nitrosamines are metabolized into N-nitrosonornicotine (NNN) and nicotine-derived nitrosamine ketone (NNK). NNK can be further metabolized into methyldiazohydroxide, pyridyl-butyl derivatives, and formaldehyde, and NNN degrades into hydroxyl or keto pyridyl-butyl [75]. In fact, e-cigarette vapor becomes toxic through degradation into potent carcinogenic nitrosamines, the induction of DNA damage in vitro [76] and subsequent tumors in various organs, as observed in in vivo animal models [77,78]. NNK and nicotine activate the serine/threonine kinase Akt taking part in key cellular processes such as glucose metabolism, cell cycle progression and apoptosis and were found to stimulate Akt signal transduction in normal human bronchial epithelial cells [NHBE cells] and small airway epithelial cells, which serve as precursors to squamous cell carcinomas and adenocarcinomas, respectively. Even small doses of nicotine may trigger the activation of Akt A processes, which in turn may result in increased cell growth and apoptosis [79]. Nevertheless, it is difficult to establish which effects are a result of tobacco smoking or e-cigarette smoking, particularly among smokers who switch from one to the other [80]. The respiratory system is extremely sensitive to foreign bodies, such as pathogens or chemical compounds. The inhalation of potentially harmful substances can be countered by responses such as coughing and sneezing, and the presence of irritant compounds in the airway can stimulate the secretion of profuse amounts of mucus, and induce pain, as protective responses. Inhaled substances may injure the pulmonary epithelium, leading to a wide range of respiratory tract disorders [64]. E-liquids often contain flavoring agents (1–4%), most of which include a significant number of aldehydes [81]. Although the diversity of flavorings has been increasing rapidly, the most commonly used types are menthol, diacetyl and aldehydes, 2,3-pentanedione, acetoin, 2,5-dimethypyrazine, 3,4-dimethoxybenzaldehyde, vanillin, cinnamaldehyde, 2-methoxycinnamaldehyde and maltol [30]. The most prevalent aldehydes are acetaldehyde and formaldehyde while the most prevalent nonaldehydes are acetoin and diacetyl [82]. 

Some chemicals in e-liquids exert a harmful effect on the lungs [83]. Diacetyl and 2,3-pentanedione impair cilia function in NHBE cells by downregulating the genes taking part in cilia biogenesis [84]. Recent studies have highlighted the role of flavoring agents on airway tract cells and the consequence of inhaling them on ROS release, cytotoxic response, immune-mediated responses, DNA damage and mucociliary clearance [30]. In addition, food flavorings are also commonly used as e-liquid additives, one example being diacetyl, which imparts a buttery flavor to food. However, while being safe for consumption in food, the inhalation of high doses of diacetyl was found to be associated with the incidence of bronchiolitis obliterans, or “popcorn lung”, a severe irreversible lung disease, in microwave popcorn production workers [85]. Diacetyl is present in e-liquid above the laboratory limit of detection [86]. The buttery flavor in e-liquid is also supported by the presence of acetoin, which may further contribute to the toxicity of the vapor [87]. Sweet-flavored e-liquids also often contain diacetyl and acetyl propionyl at concentrations exceeding safety limits [88]. Among the aldehyde components, acetaldehyde and formaldehyde can react with DNA, lipids and proteins and can create adducts [89]. These carbonyls are formed in significant amounts, with similar levels of formaldehyde being detected both in e-liquid vapor and tobacco smoke [23]. They may also play an important role in the progression of COPD and lung disease [90] and take part in the pathogenesis of lung cancer [91]. Particles containing formaldehyde, acetylaldehyde and ROS can form deposits in the bronchioles or alveoli and cause inflammatory damage of the pulmonary epithelium [92].

## 4. In Vitro Studies on E-Liquid Exposure to Cells

Cells have been commonly used to analyze the biological effects of e-cigarette liquid e-liquid. Such studies have assessed cell viability, DNA damage, ROS levels, inflammatory mediator production, changes in cell morphology and function, dysregulation of gene expression and pro-tumor events after vapor exposure using a variety of approaches and cell lines. E-cigarette usage has been found to trigger cell cytotoxicity and genotoxicity, and vapor exposure is also related to oxidative stress, with a strong connection being found between vapor exposure and overproduction of proinflammatory agents.

The studies performed on human epithelial normal bronchial cell (Nuli1), human premalignant dysplastic oral mucosal keratinocyte (POE9n) and human oral squamous cell carcinoma (UM-SCC-1) lines revealed that e-cigarette aerosols induce a dose-dependent and nicotine-independent increase of DNA damage. Exposure to vapor is characterized by elevated ROS level and increased presence of 8-oxo-7,8-dihydro-2′-deoxyguanosine (8-oxo-dG), the most abundant oxidative stress-induced DNA lesions [93]. 8-oxo-dG lesions are removed from DNA by base excision repair (BER) [94]. Polymorphisms in the enzyme involved in the BER process may affect the efficiency of repair [95]; in addition, chronic oxidative stress may induce mutations in tumor suppressor genes such as p53 [96] and proto-oncogenes such as Ras [97]. This combination of impaired BER activity and accumulated DNA damage is associated with increased cancer risk. 

Additionally, a strong connection has been reported between vapor exposure and inflammation. Lerner found exposure to e-cigarette aerosol secretion by airway epithelial cells (H292) and human fetal lung fibroblasts (HFL1) increased the levels of IL-6 and IL-8 and exhibited stress followed by morphological changes [98]. Elsewhere, exposure to vapor has been found to accelerate the cytotoxic effect and stimulate ROS and inflammatory mediator production, thus triggering the inflammatory state, in human alveolar macrophages obtained from never-smokers [99]; in addition, normal human bronchial epithelial (NHBE) cells collected from a human donor exposed to vaporized e-liquid containing different concentrations of nicotine displayed excessive cytokine production [100,101]. Inflammation can act synergistically with DNA damage to induce mutations [102] which in turn may give rise to altered proteins, thus hastening the induction of human cancer [103]. E-cigarettes have also been found to induce adverse effects on the metabolomic status of NHBEcells comparable to those seen with tobacco smoke, with these being mediated by oxidative stress [104].

The treatment of cells with e-liquid without nicotine also caused certain toxic effects; this has been attributed to the presence of flavoring chemicals. The data indicate that diacetyl, cinnamaldehyde, acetoin, pentanedione, o-vanillin, maltol and coumarin, administered at different concentrations, trigger an inflammatory response and IL-8 secretion mediated by ROS production [105]. Additionally, cinnamaldehyde has been found to impair mitochondrial function and reduce ATP levels, which correlates with the suppression of ciliary beat frequency [106]. Exposure to e-cigarette vapor was found to induce significant phenotypical changes in the immortalized NHBE-CL1548 cell line [107]. 

In contrast, inhalation of flavorless nicotine-containing e-cigarette vapor induces several adverse effects such as dysregulation in the oxidative stress response [108] and the phospholipid and fatty acid triacylglycerol metabolism pathways [109]; it has also been connected with gene dysregulation, DNA damage, apoptosis [110] and inflammatory protein release [100,111]. Nicotine inhalation has profound effects upon mitochondrial [112] and lung functions by the dysregulation of fluid homeostasis [113] and mucociliary function [114,115]. Exposure to nicotine-containing e-cigarette fluids has been found to increase mutational susceptibility [75] and phenotypical changes [107]. Moreover, the presence of nicotine in e-cigarette vapor mostly elicits a cellular response. It would appear that the inhalation of nicotine-containing vapor during e-cigarette use may have potentially dangerous effects.

While most in vitro studies suggest that e-cigarettes have a negative impact on cells, the body of evidence indicates that e-cigarette aerosol has a limited mutagenic effect on transgenic mouse fibroblast or human fibroblast cells [116]. A study conducted on human lung epithelial cells (A549) exposed to 0–20 mg/mL e-liquid for 24 h found no cytotoxic, genotoxic or inflammatory effects [117]. 

Similarly, there is a lack of consensus on the possible harmful effects of e-cigarette vapor on a 3D differentiated co-culture model of human airways. The study, conducted on a multicellular model of human cells comprising human bronchial epithelial cells (CALU-3) and pulmonary fibroblasts (MRC-5), indicate that exposure to e-liquid ingredients including PG:VG, 16 mg/mL nicotine and proprietary flavors significantly impact cell viability, proinflammatory mediator production and oxidative stress [118]. In turn, a study on EpiAirway tissue, a human airway culture derived from primary human tracheal/bronchial epithelial cells, demonstrated similar cell viability to untreated controls following exposure to commercially available e-cigarette aerosols, with and without menthol flavoring, containing 4.5% and 3% nicotine by volume [119]. 

The adverse impact of exposure to e-cigarette vapor on different cell lines is summarized in Table 1.

## 5. In Vivo Studies on E-Cigarette Liquid Exposure in Animal Models

Most animal models used for testing have been based on mice, but some rat models have been used. The animals are typically exposed to an e-cigarette liquid consisting of a base containing various concentrations of nicotine and different types of flavors. The following indicators have been determined after vapor exposure: respiratory system mechanics, mitochondrial function, oxidative stress, DNA damage, inflammation, changes in tissue or organ morphology and function, behavioral changes, dysregulation of gene expression, changes in maternal and offspring metabolism and pro-tumor events. It was observed that e-cigarette liquid alters the physiology of the lung, induces mitochondrial dysfunction and can elevate oxidative stress. E-cigarette vapor has a negative effect on the health of the animal, induces DNA damage and can trigger inflammatory responses. Therefore, it is suggested that vaping has a harmful impact on the functioning of the heart and liver. The mice have also been found to demonstrate changes in behavior and gene expression following after exposure to e-liquid vapor. Maternal e-cigarette use during pregnancy was found to have a significant impact on offspring health. In addition, collected data indicates that inhaled e-liquid enhances various types of cancer event. 

In an in vivo study, C57BL6 male mice were exposed to air, cigarette smoke and e-cigarette vapor consisting of PG:VG or PG:VG with 18 mg/mL nicotine (PG:VG-N) or PG:VG-N with flavor (PG:VG-N+F). The exposure time was set at three days or four weeks; after which, oxidative stress, inflammatory markers and pulmonary mechanics were measured. It was observed that PG:VG-N+F increased bronchoalveolar lavage fluid (BALF) cellularity and Muc5ac production, and BALF and lung oxidative stress markers were found to be at a comparable or even higher level than in the case of cigarette smoke. BALF composition correlates with lung function parameters, while the production of one of the mucins, Muc5ac, indicates airway hyperreactivity. The results clearly indicate that e-cigarette vapor has a negative effect on respiratory system mechanisms, can trigger inflammatory responses and that both e-cigarette vaping and cigarette smoking have a harmful impact on lung function [140]. 

An evaluation of the biological impact of the nicotine delivery system on the lungs of mice exposed to vapor for one hour daily for four months found that inhaled nicotine enhances airspace enlargement, mucous cell hypertrophy and the release of inflammatory mediators, processes correlated with the pathogenesis and progression of COPD [100]. In other studies, mice exposed to e-cigarette aerosols demonstrated increased proinflammatory cytokine levels [98], as well as excessive production of ROS and airway inflammation [141]. E-cigarette vapor exposure was associated with a significant increase of cytochrome P450 enzyme activity, considerable amounts of free-radicals in the lungs, reduced antioxidant capacity and extensive DNA damage in a rat lung model [142]. Salturk et al. report hyperplasia and metaplasia of the laryngeal mucosa of Wistar albino rats following exposure; however, these findings were not statistically significant [143]. In addition, exposure to e-cigarette vapor has been linked with disturbed lung development [144] and cognitive and epigenetic changes in neonatal mice [145]. It appears that e-cigarette exposure influences the overall homeostasis of the organism by altering the physiology of the lung. It is not recommended that e-cigarettes be used during pregnancy. 

Although some e-liquids do not contain nicotine, all those used in the selected in vivo studies did, and the nicotine contents of e-cigarettes are variable. Flavorless e-liquids were used for many of these studies (Table 2).

The adverse impact of exposure to e-cigarette vapor in animals is summarized in Table 2.

## 6. In Vivo Studies on E-Cigarette Liquid Exposure on Human Organisms

The effects of e-cigarette on human organisms are largely unexplored. It is difficult to establish which effects are the results of conventional smoking and which are those triggered by e-cigarette use [64]. The literature on the subject both confirms that e-cigarette aerosol has harmful effects on human health while also indicating potential benefits. 

Exposure to e-cigarette vapor in human model may impact pulmonary and vascular functions, dysregulate lung homeostasis, increase cardiac risk, and elevated oxidative stress and inflammation. It has also been found that gene expression and molecular pathways regulation were impaired by e-cigarette use. 

It appears that short term-exposure to e-cigarette vapor in human model may increase total respiratory impedance, flow respiratory resistance and peripheral airway resistance, and decrease the level of exhaled nitric oxide. E-cigarettes were found to exert a harmful physiological effect on the respiratory system comparable to that of tobacco smoking in an experimental group of healthy smokers [175]. Another study also found that parameters of lung function were impaired in e-cigarette users as compared to a control group of healthy adults [176]. Exposure to e-cigarette vapor has also been found to lead to elevated levels of oxidative stress and inflammation [177] as well as increased cardiac risk [178,179]. Thus, e-cigarette inhalation appears to have the potential to drive lung and vascular pathologies. It is important to note that the concentration of nicotine and the presence of flavors are not given in all studies reviewed herein. However, on the other hand, findings suggest that some e-cigarette components are of good quality. PG/VG and nicotine are mostly of pharmaceutical-grade quality, and the flavorings are the same as those used in food products. Additionally, the emission of acrolein and nitrosamines is significantly lower than in cigarette smoke, metals are present only in trace amounts, acetaldehyde and formaldehyde levels are lower than in the case of exposure to other air pollutants, and solid particles or carbon monoxide are absent [180]. A 42-month observation of e-cigarette users who had never smoked tobacco provided no evidence for impaired lung and heart parameters. It is again important to note that e-cigarettes are not intended as a product for nonsmokers; rather, they can offer some beneficial effects and protect against the detrimental consequences of tobacco smoke for ex-smokers. In addition, their potential to reduce the urge to smoke allows them to offer effective support in reducing nicotine usage and enhance smoking cessation [181]. E-cigarette use has greater success rates than nicotine replacement therapy [182], with many ex-smokers demonstrating decreased biomarkers for toxicity and an improvement in several clinical parameters following the switch from smoking to vaping [183,184]. Three-year studies on COPD patients indicate that e-cigarette use mostly brings benefits or even reverses some harmful effects of cigarette smoking on health: switching to e-cigarettes has been associated with exacerbations being reduced by half and deterioration in respiratory physiology being prevented [185]. In patients with asthma, significant improvements of lung function have also been observed [186]. 

The adverse impact of e-cigarettes vapor exposure on humans is summarized in Table 3.

## 7. E-Cigarette Vapor vs. Cigarette Smoke

Tobacco use kills more than eight million people worldwide each year; this number comprises about seven million as a result of direct tobacco use and about 1.2 million passive smokers [192]. Cigarette smoking causes numerous adverse health effects which can be organized in order of time of appearance, i.e., immediate effects, intermediate and long-term effects. The immediate effects include physiological markers of diminished health status, such as increased oxidative stress, depletion of circulating antioxidant micronutrient concentrations, increased inflammation, impaired immune status and altered lipid profiles, in addition to poorer self-rated health status, respiratory symptoms such as coughing, phlegm production, wheezing and dyspnea, as well as nicotine addiction. The intermediate-term effects include absenteeism, increased utilization of medical services, subclinical atherosclerosis, impaired lung development and accelerated decline in function, increased susceptibility to infectious lung diseases, diabetes, periodontitis and asthma exacerbation, as well as a higher chance of various adverse surgical outcomes: wound healing and respiratory complications. Finally, the long-term morbidity effects include cancer, vascular disease, COPD, eye diseases such as age-related macular degeneration and nuclear cataracts, rheumatoid arthritis, as well as impaired bone health, such as a greater chance of hip fractures and reduced bone density [193]. 

Although e-cigarettes are considered as alternatives to conventional tobacco products, the rapid increase in the use of electronic nicotine delivery systems has brought the need to analyze the effects of nicotine addiction and vaping-related disease [194]. Regarding e-cigarette use, the World Health Organization (WHO) has advised that products containing nicotine are unsafe for young people and pregnant women, exposure to e-liquid may increase the incidence of cancer, cardiovascular and pulmonary disease, and bystanders are exposed to nicotine and toxic substances exhaled by e-cigarette users [192]. Although vaping offers benefits for former smokers, the long-term effects of e-cigarette use remain unclear, and further research is needed to understand them better.

Some studies assess the harmfulness of exposure to e-cigarette liquid in relation to traditional cigarette smoke. One such study has found that exposure to e-cigarette aerosol elicited significantly less cytotoxicity among NHBE cells compared to tobacco smoke [195,196] and no cellular oxidative stress responses [197]. Another study also found 21 e-liquids to be less cytotoxic against cultured myocardial cells than tobacco smoke [198]. In addition, while exposure to cigarette smoke was found to be closely connected with dysregulated expression of numerous genes in NHBE cells, including those encoding proteins taking part in oxidative stress and cell death processes, exposure to e-cigarette vapor elicited minimal modulation of gene expression [199]. Additionally, such findings indicating the lower harm of e-cigarette aerosol has been confirmed in human respiratory 3D tissue models [200]. Another study on human volunteers suggests that e-cigarette smoking is significantly less harmful to lung physiology and may be recommended for tobacco smoking cessation [201]. Interestingly, the results of the in vitro cell transformation assays, useful in assessing the carcinogenic activity of particular chemicals and complex mixtures, suggest that e-cigarettes are a better alternative to traditional cigarettes [202], the compounds in the aerosol may potentiate the genotoxic effects of tobacco carcinogens when used simultaneously with conventional smoke [203]. One such study conducted on rats demonstrates that e-cigarette exposure has a detrimental effect on pulmonary structures comparable to tobacco smoke [204].

Briefly, studies comparing exposure to e-cigarettes vs air control, as in most studies described above, indicate a negative result for vaping, while those comparing them with tobacco smoke give a positive result. 

## 8. Environmental Impact of E-Cigarettes

The impact of electronic cigarettes on public health includes a range of consequences for the environment, such as air quality effects, energy and materials used, issues related to environmentally responsible disposal and land-use decisions.

It has been estimated that cigarette smoke comprises 98 chemicals constituents listed as hazardous [205], and their impact on human health and the environment is well documented [193,206,207,208]. Many of the toxic and carcinogenic agents in tobacco cigarette smoke, including polycyclic aromatic hydrocarbons, volatile organic compounds (VOCs), nitrosamines and carbon monoxide, arise as byproducts of combustion [209]. Because e-cigarettes do not have a combustion source, the health risks of vaping are believed to be significantly less than those associated with traditional cigarette smoking. E-cigarette vapor has been found to be less complex than cigarette smoke, with 16 chemicals being present at detectable levels: formaldehyde, acetaldehyde, acrolein, allyl alcohol, glyoxal, methylglyoxal, glycerol, propylene glycerol, chrysene, nicotine, myosmine, cotinine, NNN, chromium, acetone, butyraldehyde; in addition, the vapor typically contains chemicals with lower toxicities, and the emission of hazardous substances was found to be 82% to >99% lower [210].

On the one hand, research indicates that passive exposure to aerosol is not benign. are One of its main air pollutants is airborne particulate matter (PM), comprising small particles consisting of solid and liquid droplets; of these, the most adverse health effects are associated with the inhalation of those of the PM10 size fraction and fractions with lower aerodynamic diameters. It has been found that during use, e-cigarettes produce PMs and moreover PM10 was almost made of PM1 size fraction [3]. There confirm other findings suggesting that e-cigarettes emit PM10, VOCs, nicotine and definitely impair air quality [211]. Analyses of the health risk of e-cigarette use for bystanders, i.e., passive smoking, indicate that exposure may be associated with irritation of the respiratory tract, systemic effects of nicotine action and increased risk of tumors [212].

On the other hand, studies suggest that e-cigarette emission is composed mainly of PG and/or VG. Moreover, the emission profiles of the components differ between e-cigarettes and conventional cigarettes, with respective lifetimes of approximately 10–20 s and 1.4 h [213]. Exhaled particles and liquid droplets produced by e-cigarettes are smaller than for conventional tobacco and evaporate almost immediately after exhalation, unlike for conventional cigarettes [214]. Particularly noteworthy here is the problem of vaping in small rooms, in close proximity to other people, and around children [215]. 

Data gained by analyzing the environmental impact of e-cigarettes will serve to identify current and future environmental implications associated with the production, use and disposal of e-cigarettes; this will clearly be of value for regulatory authorities, e-cigarette producers and the public [216]. 

## 9. Conclusions

The present paper reviews the mechanisms of e-cigarette action, effects of exposure on e-liquid, the impact on living organisms and the environment. It summarizes evidence obtained from cell cultures, animal models and human subjects, compares vaping with smoking and assesses the impact of e-cigarettes on air quality as an environmental aspect connected with human health. Further studies should examine the effects of aerosols on living organisms more thoroughly; only relatively few studies have so far been conducted, particularly on humans, and the findings remain unclear and need further analysis. As the effects of e-cigarette use on health and the environment will probably only become apparent in the near future, it is important to provide accurate data to fully educate the public about their use.

## Figures and Tables

**Table 1 ijms-21-00652-t001:** The impact of e-cigarette liquid exposure to various cell models (in vitro study).

	Type of Cells	Characteristic of E-Liquid	Action	Ref.
1.	NHBE cells	Base: PG:VG (50%/50%) Nicotine: 36 mg/mL Flavor: flavorless	Decreased: airway surface liquid hydration Increased: mucus viscosity	[114]
2.	NHBE cells	Base: PG:VG Nicotine: 24 mg/mL Flavor: “Tennessee Cured”	Decreased: cell viability Increased: oxidative stress	[120]
3.	NHBE cells	Base: PG:VG (55%/45%) Nicotine: none Flavors: three cinnamon flavors	Decreased: intracellular ATP levels Impaired: mitochondrial respiration and glycolysis	[106]
4.	NHBE cells	Base: VG 100% Nicotine: 1.1% Flavors: tobacco	Inhibited: epithelial ion transport beyond cystic fibrosis transmembrane conductance regulator	[121]
5.	NHBE cells	Base: unknown Nicotine 2.4% w/v Flavor: flavorless	Dysregulation of gene expression	[108]
6.	NHBE cells	Base: unknown Nicotine: 24 mg per cartridge Flavors: menthol, tobacco	Decreased: expression of genes involved in cilia assembly and movement Increased: expression of genes involved in oxidative and xenobiotic stress pathways and a marker of reactive oxygen species production	[122]
7.	NHBE cells	Base: unknown Nicotine: 36 mg/mL Flavor: flavorless	Decreased: expression of FOXJ1 and KCNMA1 Increased: interleukin (IL)-6 and IL-8 release Impaired ciliary beat frequency, airway surface liquid volume, cystic fibrosis transmembrane regulator and ATP-stimulated K+ ion conductance	[100]
8.	NHBE cells	Base: unknown Nicotine: 16 mg/mL Flavor: flavorless	Alterations in cellular glycerophopholipid biosynthesis, cytochrome P450 function, retinoid metabolism, and nicotine catabolism	[109]
9.	NHBE cells	E-cigarette liquid diluted to 100 μM by nicotine	Alters the metabolome	[104]
10.	NHBE cells THP-1 macrophages	Base: PG:VG (70%:30%) Nicotine: 18 mg/mL Flavors: three apple flavors	Decreased: efferocytosis, TNF-α, IL-6, IP-10, MIP-1α and MIP-1β release Increased: necrosis and apoptosis	[123]
11.	NHBE cells	Base: PG:VG (50%/50%) Nicotine: 24 mg/mL Flavor: flavorless	Increased: IL-8 release in response to infection	[111]
12.	NHBE cells—H292 cell line	Base: PG 100% Nicotine: 24 mg/mL Flavors: tobacco, piña colada, menthol, coffee and strawberry	Decreased: metabolic activity and cell viability Increased: interleukin [IL]-1β, IL-6, IL-10, CXCL1, CXCL2 and CXCL10 release	[124]
13.	Immortalized NHBE cell line—CL-1548	Base: unknown Nicotine: none Flavor: flavorless	Decreased: ciliated, mucus-producing and club cells [phenotypic changes]	[107]
14.	NHBE 3D cell cultures	Base: unknown Nicotine: 7μg/mL equi- nicotine units Flavor: flavorless	Decreased: cystic fibrosis transmembrane conductance regulator and the epithelial sodium channel function, which regulate fluid homeostasis in the lung	[115]
15.	Human lung carcinoma A549 cells	Base: PG:VG (50%/50%) VG 100%, PG 100% Nicotine: 18 mg/mL Flavor: flavorless	Decreased: cell viability	[113]
16.	A549 cells	Base: unknown Nicotine: 18 mg/mL Flavor: strawberry	Decreased: cell viability	[125]
17.	A549 cells	Base: unknown Nicotine: 24 mg/mL Flavor: tobacco	Increased: pneumococcal adhesion to airway cells	[126]
18.	Human lung adenocarcinoma cells—A549 and NCI-H441	Base: PG:VG (50%/50%) Nicotine: 18 mg/mL Flavors: cinnamon, tobacco, menthol	Decreased: cell viability Increased: IL-8 release	[127]
19.	Human-derived bronchial epithelial cell lines—BEAS-2B, IB3-1, C38 and CALU-3; Macrophage cell lines—J774, a mouse macrophage and THP-1 a monocyte-derived macrophage; human-derived fibroblast cell line—Wi-38	Base: PG:VG Nicotine: 0.8–16 mg/mL Flavors: flavorless, cherry, tobacco, crisp mint, menthol, apple, coffee, vanilla, strawberry	Decreased: cell viability	[128]
20.	Human-induced pluripotent stem cell-derived endothelial cells [iPSC-ECs]	Base: PG:VG (50%/50%, 80%/20%), VG 100% Nicotine: 6 and 18 mg/mL Flavors: sweet tobacco with undertones of caramel, vanilla, tobacco, sweet, cinnamon, menthol	Decreased: cell viability Increased: ROS levels, caspase 3/7 activity Promoted: low-density lipoprotein uptake, activation of oxidative stress-related pathway, impaired tube formation and migration	[129]
21.	Human pulmonary fibroblasts [hPFs]; A549 lung epithelial cells; pluripotent human embryonic stem cells [hESCs]	Base: PG:VG, VG, unknown Nicotine: 6–24 mg/mL Flavors: fruit, tobacco, mint, chocolate, vanilla, caramel, candy, coffee, cinnamon, creamy	Decreased: cell viability	[130]
22.	Human airway epithelium cells	Base: PG:VG Nicotine: 2.4% Flavor: tobacco	Morphologic differences in secretory function	[131]
23.	Lung epithelial cell line [CALU3]	Base: PG:VG (70%/30%, 55%/45%) Nicotine: 12 mg/mL Flavors: Captain Black Cigar, Peanut Butter Cookie, T-bone, Popcorn, Black Licorice, Energon [orange energy drink], Vanilla Tobacco, Banana Pudding [Southern Style], Kola, Hot Cinnamon Candies, Menthol Tobacco, Solid Menthol, Peanut Butter Cookies, Banana Pudding, and Hot Cinnamon Candies	Decreased: cell viability	[132]
24.	Human umbilical vein endothelial cells [HUVECs]	Base: PG:VG Nicotine: 6–24 mg/mL Flavors: tobacco, menthol, fruit	Decreased: cell viability Alterations in cell morphology	[133]
25.	Human lung fibroblasts [HFL-1]	Base: unknown Nicotine: 16 mg/mL Flavor: tobacco	Decreased: stability of electron transport chain complex IV subunit Increased: levels of mitochondrial ROS, nuclear DNA fragmentation, IL-8 and IL-6 release	[134]
26.	Neutrophils	Base: unknown Nicotine: 16, 24 mg/mL Flavor: tobacco	Increased: CD11b and CD66b expression, MMP-9 and CXCL8 release, NE and MMP-9 activity, p38 MAPK activation	[135]
27.	Human bronchial airway epithelial cells [H292]; human fetal lung fibroblasts [HFL1]	Base: PG:VG Nicotine: 16 mg/mL Flavors: tobacco, menthol fruit, cinnamon, candy	Decreased: cell viafbility Increased: IL-6 and IL-8 release, stress and morphological change	[98]
28.	Human Periodontal Ligament Fibroblasts	Base: PG 100% Nicotine: final concentration 10 µg/mL Flavors: hazelnut, lime, menthol	Decreased: cell viability	[136]
29.	ATII cells	Base: unknown Nicotine: 24 mg/mL Flavor: flavorless	Increased: IL-8 levels, DNA damage and apoptosis	[110]
30.	Mouse vascular endothelial cell line bEnd.3; Mouse primary brain microvascular endothelial cells	Base: unknown Nicotine: 24 mg/mL Flavor: flavorless	Increased: mitochondrial depolarization, transmembrane iron exporter Slc40a1 (crucial to maintain cellular iron and redox homeostasis) and porphyrin importer Abcb6 (linked to accelerated atherosclerosis)	[112]
31.	NHBR cell line BEAS-2B; human bronchial urothelial cell line UROtsa	Base; PG:VG (50%/50%) Nicotine: 24 mg/mL Flavor: flavorless	Increased: mutational susceptibility and tumorigenic transformation	[75]
32.	Human epithelial cells—human keratinocytes (HaCaTs); Human lung alveolar type II epithelial cells (A549 cells); Alveolar macrophages (AMs) Human polymorphonuclear leukocytes (hPMNs)	Base: PG:VG, PG, VG Nicotine: 8, 24 mg/mL Flavors: flavorless, Treasury, Highlander Grog, California Blues, Pure smoke	Decreased: antimicrobial activity against *Staphylococcus aureus*	[137]
33.	Normal epithelial cells (HaCat–a spontaneously transformed immortal keratinocyte cell line); Head and neck squamous cell carcinoma (HNSCC) cell lines (UMSCC10B—derived from a metastatic lymph node, HN30—derived from a primary laryngeal tumor)	Base: PG:VG (70%/30%) Nicotine: 12 mg/mL Flavors: “Classic Tobacco”, “Red American Tobacco”	Decreased: cell viability Increased: apoptosis and necrosis, DNA strand breaks	[138]
34.	Human osteosarcoma cell lines Saos-2 and MG-63	Base: PG/VG (50%/50%) Nicotine: 24 mg/mL Flavors: flavorless, watermelon, mango, mixed fruits, coffee, apple pie, menthol and watermelon, menthol, hot cinnamon, menthol and cinnamon	Decreased: cell viability Increased: collagen type I protein expression	[139]
35.	Normal human tracheobronchial epithelial cells (NHTE cells)	Base: unknown Nicotine: 18 mg/mL Flavor: tobacco	Increased: IL-6 release Promoted: human rhinovirus (HRV) infection	[101]
36.	Monocytic cells from human pleural tissue—U937; Human monocyte-macrophage cell line (mature monocytes-macrophages)—Mono Mac 6	Base: unknown Nicotine: none Flavors: Strawberry Zing, Café Latte, Pineapple Coconut, Cinnamon Roll, Fruit Swirl, Mega Melons, Mystery Mix (menthol flavor), American Tobacco, Grape Vape, Very Berry, and Mixed Flavors	Decreased: cell viability Increased: IL-8 secretion, oxidative stress (ROS level)	[105]
37.	Alveolar macrophages; THP-1 macrophages	Base: PG:VG (50%:50%) Nicotine: 36 mg/mL Flavor: flavorless	Decreased: cell viability	[99]

**Table 2 ijms-21-00652-t002:** The impact of e-cigarette liquid exposure on animal models (in vivo studies).

	Animals	Characteristic of E-Liquid	Action	Ref.
1.	C57BL/6J mice	Base: PG:VG Nicotine: 33 mg/mL Flavor: flavorless	Decreased: innate immunity against viral pathogens in resident macrophages Increased: surfactant-associated phospholipids in the airway, lipid deposition Altered: lung lipid homeostasis in alveolar macrophages and epithelial cells, phospholipids in alveolar macrophages	[146]
2.	C57BL/6 mice	Base: PG:VG (50%/50%) Nicotine: 18 mg/mL Flavor: tobacco	Increased: bronchoalveolar lavage fluid (BALF) cellularity, Muc5ac production, BALF and lung oxidative stress markers	[140]
3.	C57BL/6 mice	Base: unknown Nicotine: 13 mg/mL Flavor: tobacco	Decrease: hippocampal gene expression of Ngfr and Bdnf, serum levels of cytokines IL-1β, IL-2, and IL-6 Increased: expression of Iba-1, a specific marker of microglia, in the cornus ammonis 1 region of the hippocampus	[147]
4.	C57BL/6 mice	Base: PG:VG (50%/50%) Nicotine: 24 mg/mL Flavor: flavorless	Decreased: weight gain Increased: angiogenesis in mouse heart tissue, endothelial cell marker CD31 and CD34, collagen content in heart tissue	[148]
5.	C57BL/6J mice	Base: unknown Nicotine: 18 mg/mL Flavor: strawberry	Increased: IL-1β release	[125]
6.	C57BL/6J mice	Base: PG:VG Nicotine: 16 mg/mL Flavors: tobacco, menthol fruit, cinnamon, candy	Decreased: lung glutathione levels Increased: proinflammatory cytokines	[98]
7.	C57BL/6J mice	Base: PG 00% Nicotine: 2.4% Flavor: flavorless	Increased: activity in the zero maze and open field tests	[149]
8.	C57BL/6J mice	Base: PG:VG (70%/30%) Nicotine: 18 mg/mL Flavor: menthol	Increased: platelets hyperactivation with enhanced aggregation, dense and α granule secretion, activation of the αIIbβ3 integrin, phosphatidylserine expression, and Akt and ERK activation Decreased: thrombosis occlusion and bleeding times	[150]
9.	C57BL/6 mice	Base: PG:VG Nicotine: 4% by weight Flavors: different (nonmenthol) flavor mixtures	Altered: minimal squamous metaplasia in laryngeal epiglottis, and histiocytic infiltrates in the lung, genes expression	[151]
10.	C57BL/6 mice	Base: PG:VG (50%/50%) Nicotine: 24 mg/mL Flavors: flavorless	Decreased: dopamine concentration in the striatum and GABA in frontal cortex Increased: glutamate concentration in the striatum and glutamine in the frontal cortex and striatum	[152]
11.	C57BL/6 mice	Base: unknown Nicotine: 18 mg/mL Flavor: cappuccino	Increased: arterial stiffness Decreased: the maximal aortic relaxation to methacholine	[153]
12.	C57BL/6J mice	Base: unknown Nicotine: 1.8% Flavor: menthol	Decreased: pulmonary bacterial clearance, phagocytosis by alveolar macrophages Increased: oxidative stress, intranasal infection with *Streptococcus pneumonia*, lung viral titers and virus-induced illness and mortality in response to Influenza A virus infection	[141]
13.	Pregnant C57BL/6J mice	Base: PG:VG (55%/45%) Nicotine: 24 mg/mL Flavor: flavorless	Impair embryo implantation Altered: integrin, chemokine, and JAK signaling pathways	[154]
14.	Pregnant C57BL/6 mice	Base: unknown Nicotine: 13–16 mg/mL Flavor: tobacco	Alerted: transcriptome in frontal cortex in both offspring and treatment groups	[155]
15.	Neonatal C57BL/6J mice	Base: PG 100% Nicotine: 1.8% Flavor: flavorless	Decreased: weight gain Increased: cotinine levels Altered: lung growth	[144]
16.	C57BL/6 mice; CD-1 mice	Base: PG:VG (50%/50%) Nicotine: 24 mg/mL Flavor: flavorless	Decreased: renal filtration, heart rate Increased: circulating proinflammatory and pro-fibrotic proteins, fibrosis in kidneys, heart and liver, blood pressure Altered: gene expression–activation of pro-fibrotic pathway, cardiovascular function	[111]
17.	Balb/c mice	Base: PG:VG Nicotine: 12 mg/mL Flavor: tobacco	Decrease: parenchymal lung function at both functional residual capacity and high transrespiratory pressures	[156]
18.	Balb/c mice	Base: unknow Nicotine: 16.8 mg/day Flavor: flavorless	Increased: brain cotinine and nicotine levels, urine cotinine levels, α4β2 nicotinic acetylcholine receptors in different brain areas	[157]
19.	Balb/c mice	Base: unknown Nicotine: 16 mg/mL Flavors: flavorless	Decreased: asthmatic airway inflammation and airway hyperresponsiveness Increased: infiltration of inflammatory cells into airways from blood, cytokines such as interleukin (IL)-4, IL-5 and IL-13 release, and ovalbumin -specific IgE production	[32]
20.	Balb/c mice	Base: PG:VG (50%/50%) Nicotine: 12 mg/mL Flavor: Black Licorice, Kola, Banana Pudding, Cinnacide	Decreased: airway inflammation Increased: peripheral airway hyperresponsiveness, soluble lung collagen Heterogeneous effects on features of allergic airways disease	[158]
21.	Pregnant Balb/C mice	Bae: unknown Nicotine: 18 mg/mL Flavor: tobacco	Decrease: global DNA methylation, Aurora Kinase (Aurk) A and AurkB gene expression and a reduction in neuronal cell numbers in the cornu ammonis 1 region of the dorsal hippocampus in offspring from mothers switching to e-cigarettes	[159]
22.	Pregnant Balb/c mice	Base: PG:VG (50%/50%) Nicotine: 18 mg/mL Flavor: tobacco	Increased: markers of oxidative stress, inflammation, and fibrosis in the adult offspring	[160]
23.	Pregnant Balb/c mice	Base: unknown Nicotine: 18 mg/mL Flavor: flavorless	Decreased: body fat in offspring Increased: mRNA expression of brain NPY and iNOS in offspring	[161]
24.	Pregnant Balb/C mice	Base: PG:VG (50%/50%) Nicotine: 18 mg/mL Flavor: tobacco	Decreased: anxiety, and hyperactivity Increased: global DNA methylation in the brains Altered: genes associated with modulating neurological activity Deficits in short-term memory in offspring	[145]
25.	Pregnant Balb/c mice	Base: PG:VG (50%/50%) Nicotine: 18 mg/mL Flavor: tobacco	Increased: IL-1β, IL-6, and TNF-α release in the mother lung, TNF-α protein levels in adult offspring Decreased: IL-1β level in adult offspring Altered: global DNA methylation	[162]
26.	CD1 mice	Base: unknown Nicotine: 24 mg/mL Flavor: tobacco	Increased: nasal platelet-activating factor receptor (PAFR) expression and nasopharyngeal pneumococcal colonization	[126]
27.	CD1 mice	Base: PG:VG (50%/50%) Nicotine: 24 mg/mL Flavor: flavorless	Decreased: glutamate transporter-1 expression in striatum, cystine/glutamate antiporter in striatum and hippocampus Increased: expression of alpha-7 nicotinic acetylcholine receptor (α-7 nAChR), which regulates glutamate release in frontal cortex and stratum, level of nicotine and cotinine in frontal cortex	[163]
28.	CD-1 mice	Base: PG:VG, PG 100% Nicotine: 0.6%, 1.8%, 14–24 mg/mL Flavors: flavorless, Treasury, Highlander Grog, California Blues, Pure smoke	Increased: acute phase reactant in serum Altered: inflammatory markers within the airways	[137]
29.	ICR mice	Base: VG 100% Nicotine: 0.5 and 5 mg/mL Flavor: flavorless	Decrease: in the grip strength and swimming time of the mice, glycogen storage in liver and muscle	[164]
30.	FVBN mice	Base; PG:VG (50%/50%) Nicotine: 24 mg/mL Flavor: flavorless	Decreased: DNA-repair activity and repair proteins XPC and OGG1/2 in the lung Increased: DNA damage in the lung, bladder, and heart	[75]
31.	FVB/N mice	Base: PG:VG (50%/50%) Nicotine: 36 mg/mL Flavor: flavorless	Increased: lung adenocarcinomas and bladder urothelial hyperplasia cases	[165]
32.	A/J mice	Base: PG:VG (50%/50%) Nicotine: 18 mg/mL Flavor: flavorless	Increased: airway hyper-reactivity, distal airspace enlargement, mucin production, cytokine and protease expression	[100]
33.	C57BL/6 mice; Apolipoprotein E knockout [ApoE−/−] mice	Base: unknown Nicotine: 2.4% Flavor: tobacco	Decreased: body weight, food intake Increased: Plasma nicotine and cotinine levels, locomotion	[166]
34.	Apolipoprotein E knockout [ApoE−/−] mice	Base: unknown Nicotine: 2.4% Flavor: flavorless	Decreased: left ventricular fractional shortening and ejection fraction Increased: oxidative stress, mitochondrial DNA mutations, atherosclerotic lesions Altered: genes associated with metabolism, circadian rhythm, inflammation, ultrastructural abnormalities indicative of cardiomyopathy	[167]
35.	Apolipoprotein E knockout [ApoE−/−] mice on a western diet	Base: unknown Nicotine: 2.4% Flavor: flavoreless	Increased: hepatic lipid accumulation, oxidative stress, hepatic triglyceride levels, hepatocyte apoptosis Altered: gene expression (including genes associated with lipid metabolism, cholesterol biosynthesis, and circadian rhythm)	[168]
36.	Apolipoprotein E knockout [ApoE−/−] mice	Base: unknown Nicotine: 2.4% Flavor: tobacco	Decreased: NAD+/NADH ratio, sirtuin 1 (SIRT1) Increased: DNA damage (apurinic/apyrimidinic sites), oxidative stress in hepatic cells, poly (ADP ribose) polymerase (PARP1) activity, mitochondrial DNA mutations, and PTEN-induced kinase 1 (PINK1), vacuolization of the mitochondria and a reduction in cellular organelles in hepatocytes	[169]
37.	DJ-1 knockdown mice	Base: Unknown Nicotine: 24 mg/mL Flavor: flavorless	Altered: regulation of oxidative phosphorylation complexes	[110]
38.	Sprague Dawley rats	Base: unknown Nicotine: 12 and 24 mg/mL Flavor: tobacco	Increased: necrosis in dorsal skin flaps	[170]
39.	Sprague Dawley rats	Base: PG:VG Nicotine: 18 mg/mL Flavor: red fruits	Increased: phase-I carcinogen-bioactivating enzymes activity, oxygen free radical production, DNA oxidation, DNA damage at chromosomal and gene level	[142]
40.	Wistar rats	Base: PG:VG (50%/40%) Nicotine: 18 mg/mL Flavor: tobacco	Decreased: sperm vitality, sperm count in the cauda epididymis Increased: morphologically abnormal spermatozoa, myeloperoxidase granules–inflammatory state Altered: semen parameters, redox status	[171]
41.	Wistar rats	Base: PG:VG (50%/40%) Nicotine: 18 mg/mL Flavor: tobacco	Decreased: total protein content (superoxide dismutase, catalase and glutathione-S-transferase), cell viability Increased: malondialdehyde content, lipid peroxidation, oxidative stress, inflammatory cells infiltration, oxidative tissue injuries	[172]
42.	Wistar rats	Base: PG:VG (50%/40%) Nicotine: 18 mg/mL Flavor: tobacco	Decreased: uric acid and mainly urea, superoxide dismutase and catalase activities Increased: total protein and sulfhydryl content, cells with reduced and dark nuclei located in the renal collecting ducts	[173]
43.	Wistar rats	Base: PG:VG (50%/40%) Nicotine: 18 mg/mL Flavor: tobacco	Decreased: sperm density and viability, testicular lactate dehydrogenase activity, testosterone level, cytochrome P450 side-chain cleavage, 17 beta-hydroxysteroid dehydrogenase mRNA level Increased: antioxidant enzyme activities such as superoxide dismutase, catalase and glutathione-S-transferase, sulfhydryl group content Altered: testis tissue—germ cells desquamation, disorganization of the tubular contents of testis and cell deposits in seminiferous tubules	[174]

**Table 3 ijms-21-00652-t003:** The impact of e-cigarette liquid exposure on e-cigarette users (in vivo studies).

	Patients	Characteristic of E-Liquid	Action	Ref.
1.	Healthy never-smokers (before and after e-cigarette usage)	Base: PG:VG (70%/30%) Nicotine: nicotine-free Flavor: flavorless	Increased: oxidative stress, inflammation, circulatory burden of the serum	[177]
2.	Healthy never-smokers (before and after e-cigarettes usage)	Unknown	Increased: plasma endothelial microparticle levels Altered: transcriptomes of small airway epithelium and alveolar macrophages	[187]
3.	Healthy never-smokers (before and after e-cigarettes usage)	Base: PG:VG Nicotine:1.2% Flavor: tobacco	Decrease: high-frequency spectral component in electrocardiogram (0.15-0.4 Hz) Increase: low-frequency spectral component in electrocardiogram (0.04–0.15 Hz) and the low-frequency to high-frequency ratio Altered: in cardiac sympathovagal balance towards sympathetic predominance	[178]
4.	Healthy e-cigarette users (divided into groups either exposed or not exposed to e-cigarette vapor)	Unknown	Altered: gene expression taking part in Wnt/Ca+ pathway and Rho family GTPases signaling pathway in oral cells	[188]
5.	Healthy e-cigarettes users vs controls subjects (never smokers)	Unknown	Decreased: high-frequency spectral component in electrocardiogram Increased: low-frequency spectral component in electrocardiogram, low frequency to high frequency ratio, low-density lipoprotein oxidizability, oxidative stress	[179]
6.	Healthy e-cigarettes users (before and after e-cigarettes usage)	Base: PG:VG Nicotine: 18 mg/mL Flavor: tobacco	Decreased: peak expiratory flow Increased: particularly platelet microparticles	[189]
7.	Healthy e-cigarettes users vs controls subjects (never smokers)	Unknown	Decreased: lung function test parameters	[176]
8.	Healthy sporadic smokers (before and after e-cigarettes usage)	Base: PG:VG (49.4%/44.4%) Nicotine: 19 mg/mL Flavor: flavorless	Increased: heart rate and arterial stiffness, flow resistance, blood pressure	[190]
9.	Healthy sporadic smokers (divided into two groups either exposed or not exposed to e-cigarette vapor)	Base: PG:VG (49.4%/44.4%) Nicotine: 12 mg/mL Flavor: flavorless	Increased: endothelial progenitor cells level in blood, E-selectin positive microvesicles (endothelial origin)	[191]
10.	Healthy smokers vs controls subjects (never smokers)	Base: PD Nicotine: <10% Flavor: tobacco	Decreased: fraction of exhaled nitric oxide Increased: total respiratory impedance, flow respiratory resistance, overall peripheral airway resistance	[175]

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
