# Peer review of "A Summary of In Vitro and In Vivo Studies Evaluating the Impact of E-Cigarette Exposure on Living Organisms and the Environment"

_ijms, 2020, doi:10.3390/ijms21020652_

Round 1
Reviewer 1 Report
The impact of e-cigarettes exposure on the living organisms and environment - in vitro and in vivo study.
Comments and suggestions for the authors:
The review in question needs both a language revision (in order to eliminate some misprints present in the text) and a style revision (in order to increase the readability of the paper, particularly where basic information without citation is reported).
Some items need to be revised:
par. 1 - line 39: the e-cigarette users inhale the vapor produced from the liquid, not the liquid itself. par. 2 - line 107: oxidative stress should not be considered as an example of lung pathologies. This is not a pathology, but an imbalance condition known to contribute to the pathogenesis of several health problems. par 3 - line 114: “aerosolized PG and/or VG” should be components of the vapor and not of the liquid. Aerosolized components are produced upon the activation of e-cigarettes. table 3 - lines 3, 5: People who don’t read the cited paper don’t understand that the expressions “high-frequency component” and “low-frequency component” are referred to the spectral components of an ECG. In literature, a significant contribution of e-liquid components on the variance in nicotine in the aerosol is reported. Maybe in the review should be present also a focus on this point. the meaning of Public Health that the results highlight should be studied in depth.Author Response
Dear Reviewer,
The authors are grateful to the Reviewer for very detailed analysis of our manuscript and useful suggestions. We appreciate all the insightful comments of the Reviewer and provide a point-by-point reply:
Comment 1:
“The review in question needs both a language revision (in order to eliminate some misprints present in the text) and a style revision (in order to increase the readability of the paper, particularly where basic information without citation is reported).”
Response 1:
According to the Reviewer suggestion, the English has been corrected by British native speaker.
Comment 2:
“Some items need to be revised:
par. 1 - line 39: the e-cigarette users inhale the vapor produced from the liquid, not the liquid itself. par. 2 - line 107: oxidative stress should not be considered as an example of lung pathologies. This is not a pathology, but an imbalance condition known to contribute to the pathogenesis of several health problems. par 3 - line 114: “aerosolized PG and/or VG” should be components of the vapor and not of the liquid. Aerosolized components are produced upon the activation of e-cigarettes. table 3 - lines 3, 5: People who don’t read the cited paper don’t understand that the expressions “high-frequency component” and “low-frequency component” are referred to the spectral components of an ECG. “
Response 2:
According to the Reviewer suggestions, all items were changed in manuscript.
Comment 3:
“In literature, a significant contribution of e-liquid components on the variance in nicotine in the aerosol is reported. Maybe in the review should be present also a focus on this point.”
Reply 3:
Thank you for your comment, in section 4, 5 and 6 the information about the impact of nicotine on living organisms were added.
Comment 4:
“The meaning of Public Health that the results highlight should be studied in depth.”
Reply 4:
According to the Reviewer suggestion, the information about the e-cigarettes exposure and their impact on Public Health has been described in section 7. All changed fragments have been marked in turquoise in the manuscript.
Sincerely Yours,
Anna Merecz-Sadowska

Reviewer 2 Report
General Comments:
In my opinion the title of the paper is quite confusing. It refers to an in vitro and in vivo study, whereas, this is a review manuscript of the literature. I think that authors should revise their title referring more appropriately to the purpose(s) of their manuscript. I think that in the introduction the paragraphs “Carcinogenesis…” and “Lung cancer…”, are over-analyzed. I would expect a more descriptive analysis of the e-cigarette’s devises, then maybe a description of their possible effects (or possible mechanisms of their use/action). Although the authors have a separate paragraph “The mechanisms of e-cigarette action”, this still is quite vague to me. Also, authors state at the end of the Introduction that “This review examines the mechanisms of action of e-cigarettes and the effect of exposure to e-liquid in vitro and in vivo. It also evaluates the environmental impact of electronic nicotine delivery system.” I think that the authors should state that their scope is the evaluation of the literature on these aspects. Unless this is an original article and not a review manuscript, then how they evaluate mechanisms of action….in vitro and in vivo? This comment is also related with the misleading message of the manuscript title. Moreover, to this frame, key words of the manuscript need also revision, especially the words “in vitro and in vivo study”. Section “In vitro studies on e-liquid exposure to cells” seems unrelated to the Table 1 that authors postulate. This is mostly because the authors are only commenting on 2-3 studies and then they prompt the readers to the table for the rest of the published papers. In my opinion, this is not the wright way for a review paper and is quite useless. The main purpose of a review manuscript is not to rather mention or recapitulate the already published studies, but to critically comment on these findings so as the readers can benefit from reading this review paper. So, instead of just referring to corresponding Table 1, authors should try to critically explain and comparatively analyzing the concept of published works. Likewise, “In vivo studies on e-cigarette liquid exposure in animal models” paragraph is unlinked to Table 2. Although summarizing all the published studies by a means of a table is helpful, a review manuscript should in depth analyze the findings of at least the most important published papers.Concluding remark: In my opinion, the topics of this review is quite interesting and up to date, but this manuscript needs further emphasis and in-depth analysis of at least the most important published works in the field. So, in general, for a positive recommendation for publication, I would like to see a more in depth and critical analysis of the current literature by the authors, instead of a summary of them, that adds little to the potential readers of the manuscript.
Overall Recommendation: Re-evaluation of the manuscript after Major Revision
Author Response
Dear Reviewer,
The authors are grateful to the Reviewer for very detailed analysis of our manuscript and useful suggestions. We appreciate all the insightful comments of the Reviewer and provide a point-by-point reply:
Comment 1:
“In my opinion the title of the paper is quite confusing. It refers to an in vitro and in vivo study, whereas, this is a review manuscript of the literature. I think that authors should revise their title referring more appropriately to the purpose(s) of their manuscript.”
Reply 1:
According to the Reviewer suggestion the tittle has been changed, so as not to suggest the experimental type of paper.
Comment 2:
“I think that in the introduction the paragraphs “Carcinogenesis…” and “Lung cancer…”, are over-analyzed.”
Reply 2:
According to the Reviewer suggestion Introduction has been re‐organized and shortened
Comment 3:
“I would expect a more descriptive analysis of the e-cigarette’s devises, then maybe a description of their possible effects (or possible mechanisms of their use/action). Although the authors have a separate paragraph “The mechanisms of e-cigarette action”, this still is quite vague to me.”
Reply 3:
According to the Reviewer suggestion the section 2 “The mechanisms of e-cigarette action” has been significantly extended in manuscript.
Comment 4:
“Also, authors state at the end of the Introduction that “This review examines the mechanisms of action of e-cigarettes and the effect of exposure to e-liquid in vitro and in vivo. It also evaluates the environmental impact of electronic nicotine delivery system.” I think that the authors should state that their scope is the evaluation of the literature on these aspects. Unless this is an original article and not a review manuscript, then how they evaluate mechanisms of action….in vitro and in vivo?”
Reply 4:
According to the Reviewer suggestion the sentence: “This review examines the mechanisms of action of e-cigarettes and the effect of exposure to e-liquid in vitro and in vivo. It also evaluates the environmental impact of electronic nicotine delivery system.” has been changed into: “This review examines the literatures about the mechanisms of action of e-cigarettes, the effect of exposure to e-liquid in vitro and in vivo and environmental impact of electronic nicotine delivery system.”
Comment 5:
“Moreover, to this frame, key words of the manuscript need also revision, especially the words “in vitro and in vivo study”.
Reply 5:
According to the Reviewer suggestion key words of the manuscript have been changed.
Comment 6:
“Section “In vitro studies on e-liquid exposure to cells” seems unrelated to the Table 1 that authors postulate. This is mostly because the authors are only commenting on 2-3 studies and then they prompt the readers to the table for the rest of the published papers. In my opinion, this is not the wright way for a review paper and is quite useless. The main purpose of a review manuscript is not to rather mention or recapitulate the already published studies, but to critically comment on these findings so as the readers can benefit from reading this review paper. So, instead of just referring to corresponding Table 1, authors should try to critically explain and comparatively analyzing the concept of published works. Likewise, “In vivo studies on e-cigarette liquid exposure in animal models” paragraph is unlinked to Table 2. Although summarizing all the published studies by a means of a table is helpful, a review manuscript should in depth analyze the findings of at least the most important published papers. In my opinion, the topics of this review is quite interesting and up to date, but this manuscript needs further emphasis and in-depth analysis of at least the most important published works in the field. So, in general, for a positive recommendation for publication, I would like to see a more in depth and critical analysis of the current literature by the authors, instead of a summary of them, that adds little to the potential readers of the manuscript.”
Reply 6:
According to the Reviewer suggestions the section 4, 5, 6 has been analyzed in depth, current literatures were described in detail. All changed fragments have been marked in yellow in the manuscript. Additional literature points in the field were cited. Thank you very much for your positive recommendation and we hope that all changes have been approved.
Sincerely Yours,
Anna Merecz-Sadowska

Round 2
Reviewer 2 Report
The authors have adequately positively responded to the comments and I think that they merit a positive consideration for publication of their work.